# FEDERATED TRAINING OF DUAL ENCODING MODELS ON SMALL NON-IID CLIENT DATASETS

## ABSTRACT

Dual encoding models that encode a pair of inputs are widely used for representation learning. Many approaches train dual encoding models by maximizing agreement between pairs of encodings on centralized training data. However, in many scenarios, datasets are inherently decentralized across many clients (user devices or organizations) due to privacy concerns, motivating federated learning. In this work, we focus on federated training of dual encoding models on decentralized data composed of many small, non-IID (independent and identically distributed) client datasets. We show that existing approaches that work well in centralized settings perform poorly when naively adapted to this setting using federated averaging. We observe that, we can simulate large-batch loss computation on individual clients for loss functions that are based on encoding statistics. Based on this insight, we propose a novel federated training approach, *Distributed Cross Correlation Optimization (DCCO)*, which trains dual encoding models using encoding statistics aggregated across clients, without sharing individual data samples. Our experimental results on two datasets demonstrate that the proposed DCCO approach outperforms federated variants of existing approaches by a large margin.

## 1 INTRODUCTION

Dual encoding models (see Fig. 1) are a class of models that generate a pair of encodings for a pair of inputs, either by processing both inputs using the same network or using two different networks. These models have been widely successful in self-supervised representation learning of unlabeled unimodal data (Chen et al., 2020a; Chen & He, 2021; Zbontar et al., 2021; He et al., 2020; Grill et al., 2020), and are also a natural choice for representation learning of paired multi-modal data (Jia et al., 2021; Radford et al., 2021; Bardes et al., 2022).

While several approaches exist for training dual encoding models in centralized settings (where the entire training data is present on a central server), training these models on decentralized datasets is less explored. Due to data privacy concerns, learning from decentralized datasets is becoming increasingly important. Federated learning (McMahan et al., 2017) is a widely-used approach for learning from decentralized datasets without transferring raw data to a central server. In each round of federated training, each participating client computes a model update using its local data, and then a central server aggregates the client model updates and performs a global model update.

In many real-world scenarios, individual client datasets are small and non-IID (independent and identically distributed), e.g., in *cross-device* federated settings (Kairouz et al., 2021; Wang et al., 2021). For example, in the context of mobile medical apps such as Aysa (AskAysa, 2022) and DermAssist (DermAssist, 2022), each user contributes only a few (1-3) images. Motivated by this, we focus on federated training of dual encoding models on decentralized data composed of a large number of small, non-IID client datasets.

Recently, several highly successful approaches have been proposed for training dual encoding models in centralized settings based on contrastive losses (He et al., 2020; Chen et al., 2020a;b), statistics-based losses (Zbontar et al., 2021; Bardes et al., 2022), and predictive losses combined with batch normalization (Ioffe & Szegedy, 2015) and stop-gradient operation (Grill et al., 2020; Chen & He, 2021). One way to enable federated training of dual encoding models is to adapt these existing approaches using the Federated Averaging (FedAvg) strategy of McMahan et al. (2017). As

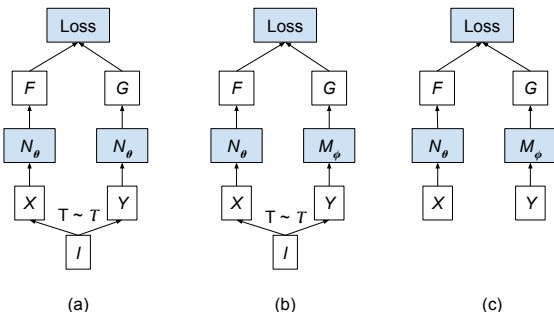

Figure 1: Dual encoding models. Two inputs generated using random data augmentations $(T \sim \mathcal{T})$ are encoded by (a) the same encoder network or (b) two different networks. In (c) aligned inputs from two different modalities are encoded by two different modality-specific networks.

described in Section 2, all of these approaches require large and diverse training batches to work well, and their performance degrades when trained on small, non-IID client datasets (see Sec. 4.4). While Zhuang et al. (2021), Makhija et al. (2022), Zhuang et al. (2022) and He et al. (2021) extend predictive loss approaches such as BYOL (Grill et al., 2020) and SimSiam (Chen & He, 2021) to federated settings, they consider only *cross-silo* settings (with thousands of samples per client) where large-batch training is possible.

In this work we observe that, in the case of statistics-based loss functions, we can simulate large-batch loss computation on each individual (small) client, by first aggregating encoding statistics from many clients and then sharing these aggregated large-batch statistics with all the clients that contributed to them. Based on this observation, we propose a novel approach, *Distributed Cross Correlation Optimization (DCCO)*, for federated training of dual encoding models on small, non-IID client datasets. The proposed approach simulates large-batch training based with the loss function of Zbontar et al. (2021), which we refer to as *Cross Correlation Optimization (CCO)* loss. This is achieved without sharing individual data samples or their encodings between clients.

Major Contributions

- We observe that large-batch training of dual encoding models can be simulated on decentralized datasets by using loss functions based on encoding statistics aggregated across clients, without sharing individual samples.

- Building on this insight, we present Distributed Cross Correlation Optimization (DCCO), a novel approach for training dual encoding models on decentralized datasets composed of a large number of small, non-IID client datasets.

- We prove that when we perform one step of local training in each federated training round, one round of DCCO training is equivalent to one step of centralized training on a large batch composed of all samples across all clients participating in the federated round.

- We evaluate the proposed DCCO approach on CIFAR-100 and dermatology datasets, and show that it outperforms FedAvg variants of contrastive and CCO training by a significant margin. The method also significantly outperforms supervised training from scratch, demonstrating its effectiveness for decentralized self-supervised learning.

## 2 Problems with Existing Approaches

**Contrastive loss functions**   Contrastive losses explicitly maximize similarity between two encodings of a data sample while pushing encodings of different samples apart. This is highly effective when each sample is contrasted against a large set of diverse samples (Chen et al., 2020a;b; He et al., 2020; Radford et al., 2021; Jia et al., 2021). Contrastive learning approaches can be extended to federated settings by combining within-client contrastive training with the FedAvg strategy of McMahan et al. (2017). But this reduces performance as each sample is contrasted against a small set of within-client samples, which may be relatively similar.

**Dependence on batch normalization** Approaches such as BYOL (Grill et al., 2020) and Sim-Siam (Chen & He, 2021) train dual encoding models using a predictive loss that encourages one encoding of a data sample to be predictive of another encoding of the same sample. Unlike contrastive losses, predictive losses do not explicitly push the encodings of different samples apart. Despite this, these approaches work well when trained with large batches in centralized settings. Importantly, both BYOL and SimSiam use batch normalization (Ioffe & Szegedy, 2015), whose efficacy decreases rapidly when batches become smaller (Ioffe, 2017; Wu & He, 2018). When training on small, non-IID client datasets in federated settings, group normalization (Wu & He, 2018) is typically used instead of batch normalization (Hsieh et al., 2020; Hsu et al., 2020). When we experimented with BYOL and SimSiam by replacing batch normalization with group normalization, the models did not train well (see Appendix C), suggesting that batch normalization is important for these approaches to work well[1]. This dependence on batch normalization suggests that these approaches are not a good fit for federated training on small, non-IID client datasets.

**Statistics-based loss functions** While the above contrastive and predictive losses directly use individual sample encodings in their computation, the CCO loss introduced by Zbontar et al. (2021) is a function of encoding statistics computed over a batch of samples. This loss function maximizes the correlation coefficient values of matching dimensions and minimizes the correlation coefficient values of non-matching dimensions of the two encodings of a dual encoding model. Since CCO loss is a function of batch statistics, its efficacy decreases with smaller batch sizes. While it works well in centralized settings where large batches are used for training, it performs poorly in federated settings when used for training on small, non-IID client datasets (see Sec. 4.4).

## 3 PROPOSED APPROACH

### 3.1 CROSS CORRELATION OPTIMIZATION LOSS

Let $X$ and $Y$ be the two inputs to a dual encoding model, and $F = [F_i] \in \mathcal{R}^d$ and $G = [G_j] \in \mathcal{R}^d$ be their encodings, respectively. The CCO loss used for training dual encoding models in Zbontar et al. (2021) is given by [2]

$$\mathcal{L}_{CCO} = \sum_{i=1}^{d}(1 - C_{ii})^2 + \lambda \sum_{i=1}^{d} \frac{1}{d-1} \sum_{\substack{j=1 \\ j \neq i}}^{d} C_{ij}^2, \tag{1}$$

where $C_{ij}$ represents the correlation coefficient between the $i^{th}$ component of $F$ and the $j^{th}$ component of $G$:

$$C_{ij} = \frac{\langle F_i G_j \rangle - \langle F_i \rangle \langle G_j \rangle}{\sqrt{\langle (F_i)^2 \rangle - \langle F_i \rangle^2} \sqrt{\langle (G_j)^2 \rangle - \langle G_j \rangle^2}}. \tag{2}$$

Here, $\langle \ \rangle$ represents average values computed using a batch of samples. The first term in the CCO loss (Eq. 1) encourages the two encodings of a data sample to be similar by maximizing the correlation coefficient between the matching dimensions of the encodings, and the second term reduces the redundancy between output units by decorrelating the different dimensions of the encodings.

### 3.2 MOTIVATION: AGGREGATING AND REDISTRIBUTING ENCODING STATISTICS

Both contrastive and CCO losses are highly effective when used with large training batches, and their efficacy decreases as batch size decreases. Contrastive losses directly use individual sample encodings, and it is unclear how large-batch contrastive loss can be computed on small clients without sharing individual sample encodings between clients. Unlike contrastive losses, CCO loss only uses encoding statistics $\langle F_i \rangle, \langle (F_i)^2 \rangle, \langle G_j \rangle, \langle (G_j)^2 \rangle, \langle F_i G_j \rangle$. Although CCO loss is a nonlinear

---

[1] Fetterman & Albrecht (2020) also observes that batch normalization is crucial for BYOL to work. Though Richemond et al. (2020) refutes this and claims that BYOL works well with group normalization, we were not successful in our attempts to train BYOL with group normalization. [2] Zbontar et al. (2021) do not refer to this loss as CCO loss; we use this more general term instead of *BarlowTwins* loss to emphasize the loss function's applicability to cases where the two encoder networks differ. We added a normalization factor $\frac{1}{d-1}$ to the loss function to reduce the sensitivity of tradeoff parameter $\lambda$ to the dimensionality of the encodings.

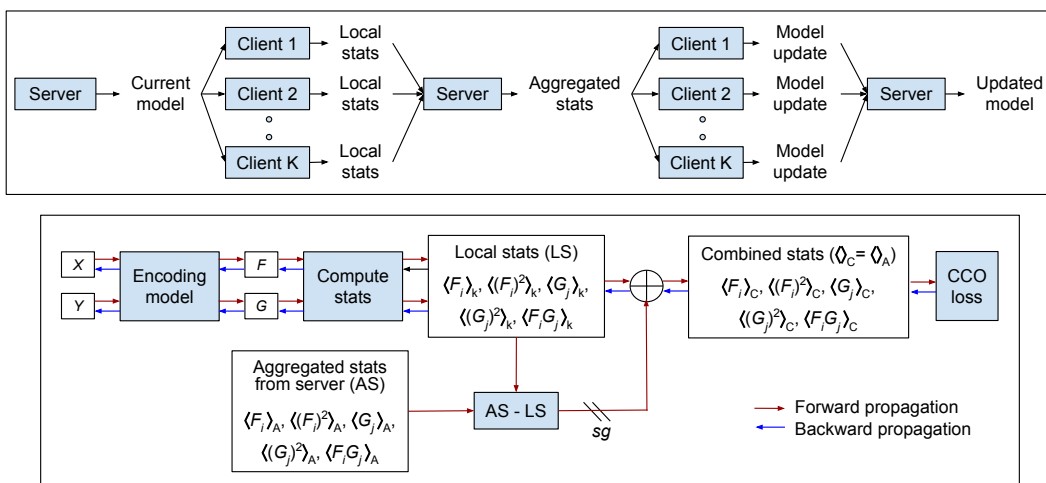

Figure 2: Top: one round of DCCO training. Bottom: local training on client $k$. Here $sg$ refers to the stop gradient operation, and $\oplus$ denotes addition. The combined statistics $\langle\rangle_C$ used for loss computation are equal to the aggregated statistics $\langle\rangle_A$ from the server. Note that gradient backpropagation happens only through the local statistics $\langle\rangle_k$.

function of these statistics, the statistics themselves are average values, i.e. linear combinations of individual sample values. Hence, we can compute large-batch statistics as a weighted average of small-batch statistics computed on individual clients, without sharing individual sample encodings. This makes it possible to simulate large-batch CCO loss on individual clients by first aggregating encoding statistics from many clients and sharing these aggregated large-batch statistics with all the clients that contributed to them. Based on this insight, we propose our DCCO approach that simulates large-batch training using small, non-IID client datasets.

The strategy of aggregating and redistributing statistics can also be used to implement multi-client synchronous batch normalization in federated settings. However, since batch normalization is typically used at every layer of the encoder network, this will add multiple additional rounds of server-client communication (as many as the number of batch normalization layers in the network) which is undesirable. In contrast, the proposed DCCO approach uses only one additional round of communication by aggregating statistics at the end of the network before computing the loss function.

## 3.3 DISTRIBUTED CROSS CORRELATION OPTIMIZATION (DCCO)

Figure 2 presents the proposed DCCO training approach. In each training round, a central server samples $K$ clients from the pool of available clients and broadcasts the current model to these clients. Each client $k$ uses this model to encode its local data and compute local encoding statistics $\langle F_i \rangle_k, \langle (F_i)^2 \rangle_k, \langle G_j \rangle_k, \langle (G_j)^2 \rangle_k, \langle F_i G_j \rangle_k$ based on $N_k$ samples. These local encoding statistics are then aggregated by the central server using weighted averaging (Eq. 3), and the aggregated statistics are shared with the participating clients.

$$\langle F_i \rangle_A = \sum_{k=1}^{K} \frac{N_k}{N} \langle F_i \rangle_k, \ \ \langle (F_i)^2 \rangle_A = \sum_{k=1}^{K} \frac{N_k}{N} \langle (F_i)^2 \rangle_k, \ \ \langle G_j \rangle_A = \sum_{k=1}^{K} \frac{N_k}{N} \langle G_j \rangle_k,$$

$$\langle (G_j)^2 \rangle_A = \sum_{k=1}^{K} \frac{N_k}{N} \langle (G_j)^2 \rangle_k, \ \ \langle F_i G_j \rangle_A = \sum_{k=1}^{K} \frac{N_k}{N} \langle F_i G_j \rangle_k, \ \ N = \sum_{k=1}^{K} N_k. \tag{3}$$

Then, each client computes a local model update by minimizing the CCO loss function computed using aggregated statistics $\langle\rangle_A$. During local training, while aggregated statistics are used for computing the loss function, gradients are backpropagated only through local statistics $\langle\rangle_k$ since each client has access only to its local data. Finally, the local model updates are aggregated by the central server and used to update the current model. To aggregate client model updates, we use a weighted

average, weighted by examples per client. During the entire training process, individual data samples never leave the clients either in raw or encoded formats.

Our work focuses on training with small client datasets (for example, no more than six images per client in the case of DERM dataset). Hence, in each federated training round, we only perform one step of local training on each participating client. In this setting, one round of DCCO training is equivalent to one centralized training step on a batch composed of all data samples across all clients participating in the federated round. See Appendix A for the proof. This equivalence between federated and centralized training holds only because of the additional step of aggregating and sharing encoding statistics. Naively applying FedAvg with one within-client CCO loss-based training step per round is not equivalent to centralized training, as we show in Section 4.

While this work focuses on the CCO loss of Zbontar et al. (2021), the proposed distributed learning strategy (statistics aggregation and redistribution followed by aggregated statistics-based local training) can also be used with other statistics-based loss functions such as Bardes et al. (2022).

## 4 EXPERIMENTS

In this section, we evaluate the proposed DCCO approach in the context of self-supervised learning where the two inputs of a dual encoding model are generated by applying random augmentations to a single unlabeled input, and both the inputs are processed using the same encoder network (see Fig. 1(a)).

**Experimental setup**   First, the encoder network is pretrained on an unlabeled, decentralized dataset. Then, a linear classifier is added on top of the encoder network, and either only the newly added classifier (linear evaluation protocol) or the entire network (full finetuning protocol) is trained using a small labeled classification dataset on a central sever. The accuracy of final model is used as the evaluation metric.

**Comparative approaches**   We compare the proposed DCCO approach with FedAvg variants of within-client contrastive and CCO loss-based training. We use centralized training based on CCO loss as an upper bound for the proposed approach. We also report results for fully-supervised training from scratch (using the limited labeled data) to demonstrate the effectiveness of DCCO as a self-supervised pretraining strategy.

### 4.1 DATASETS

**CIFAR-100 (Krizhevsky & Hinton, 2009)**   This dataset consists of 50K training and 10K test images of size $32 \times 32$ from 100 object categories. We use all 50K training images as unlabeled data for pretraining, and a small fraction of them (10% or 1%) as labeled data for supervised finetuning. To study the effect of non-identical client data distribution, we generated IID and non-IID client datasets using the Dirichlet distribution-based sampling process of Hsu et al. (2019) with $\alpha = 1000$ and $\alpha = 0$, respectively. We generated multiple decentralized datasets by varying the total number of clients and the number of samples per client (see Table 1).

**Dermatology (DERM) (Liu et al., 2020)**   This dataset consists of de-identified images of skin conditions captured using consumer-grade digital cameras. The images are heterogeneous in nature and exhibit significant variations in terms of pose, lighting, blur, and body parts. Each case in this dataset includes one to six images of a skin condition from a single patient. A portion of the cases in this dataset are labelled with 419 unique skin conditions by a panel of US board-certified dermatologists. Following Liu et al. (2020) and Azizi et al. (2021; 2022), we focus on the most common 26 skin conditions and group the rest into a common 'Other' class leading to a final label space of 27 categories. Following Azizi et al. (2022), we use a total of 207,032 unlabeled images (59,770 cases) for pretraining the encoder network. For federated pretraining, we consider each case as a separate client. For supervised finetuning, we use up to 4,592 labeled cases. The final classification accuracy is reported on a test set of 4,146 labeled cases. This dataset also provides a validation split with 1,190 labeled cases which we use for tuning some hyperparameters.

## 4.2 NETWORKS

Residual networks (He et al., 2016) with 14 and 50 layers are used as encoder networks for experimenting with CIFAR-100 and DERM datasets, respectively. Following Zbontar et al. (2021), a three layer fully-connected projection network is used to increase the dimensionality of the encodings before computing the CCO loss during pretraining, and the projection network is discarded while training the final classifier. The projection network configurations are [1024, 1024, 1024] and [2048, 2048, 4096] for CIFAR-100 and DERM datasets, respectively. For the contrastive loss, following Chen et al. (2020a;b), a fully-connected projection network is used to reduce the dimensionality of the encodings before computing the loss. The projection network configurations are [256, 256, 128] and [2048, 2048, 128] for CIFAR-100 and DERM datasets, respectively. Weight standardization (Qiao et al., 2019) and group normalization (Wu & He, 2018) with 32 groups are used at every layer except the last projection layer. For the DERM dataset, since each case consists of multiple images (up to six), the final classification network performs average pooling of individual image encodings and uses a linear classifier on top of the average-pooled feature to predict the label for a case. Following Azizi et al. (2021), we use $224 \times 224$ images as input while pretraining the encoder, and $448 \times 448$ images as input while training the classifier.

## 4.3 TRAINING DETAILS

**CIFAR-100**  We experimented with several decentralized versions of CIFAR-100 dataset by varying the total number of clients and the number of samples per client (see Table 1). During federated pretraining, all models were trained for 100K rounds. For FedAvg variants of CCO and contrastive loss-based training, we observed overfitting. For these approaches, we evaluated multiple pretrained checkpoints under linear evaluation protocol and report the results for the best checkpoints. We did not observe such overfitting for the proposed DCCO approach and report results based on 100K rounds of training. Each client is visited around 1000 times during federated pretraining of 100K rounds. So, for centralized CCO loss-based pretraining, we used 1000 epochs of training with a batch size of 512.

**DERM**  For federated experiments, each case in the dataset is considered as a client dataset. We conducted experiments by varying the number of clients sampled in each round (see Tab. 2). All models were trained for 75K rounds during federated pretraining. For this dataset (59,770 clients), each client is visited around 320 times if we perform 75K rounds of federated training with 256 clients per round. So, for centralized CCO loss-based training, we used 320 epochs of training with a batch size of 512.

We used a value of 20 for the tradeoff parameter $\lambda$ in CCO loss (Eq. 1) and a value of 0.1 for the temperature parameter in contrastive loss (Chen et al., 2020a). In all federated experiments, we used gradient descent with learning rate 1.0 as the local optimizer on clients. Please refer to Appendix B for further details about the training hyperparameters and data augmentations used in all settings.

## 4.4 RESULTS

### 4.4.1 CIFAR-100 DISCUSSION

Table 1 shows the performance of various approaches under linear evaluation and full finetuning protocols for multiple decentralized versions of CIFAR-100 dataset. When only 500 images are labeled, training only the linear classifier performed better than finetuning the entire network. So, in this case, we only report results under linear evaluation protocol. The proposed DCCO approach outperforms the FedAvg variants of contrastive and CCO loss-based training by a significant margin for both IID and non-IID client datasets. Specifically, in the case of non-IID client datasets, the performance gains are in the range of 11-18% under linear evaluation protocol and 6-10% under full finetuning protocol, and in the case of IID client datasets, the gains are in the range of 4-10% under linear evaluation protocol and 4-6% under full finetuning protocol. The proposed DCCO approach also outperforms fully-supervised training from scratch by a significant margin (10-20%) demonstrating the effectiveness of DCCO as a federated self-supervised pretraining strategy.

In the case of non-IID client datasets, for a fixed global batch size (i.e., the total number of samples participating in a round), the performance of DCCO approach increases as the number of samples

| | Non-IID client datasets | | | | IID client datasets | | |
|---|---|---|---|---|---|---|---|
| Total clients | 50,000 | 12,500 | 6,250 | 3,125 | 12,500 | 6,250 | 3,125 |
| Samples / Client | 1 | 4 | 8 | 16 | 4 | 8 | 16 |
| Clients / Round | 512 | 128 | 64 | 32 | 128 | 64 | 32 |
| Linear evaluation protocol (5K labeled training samples) | | | | | | | |
| CCO + FedAvg | – | Failed | 28.7 | 30.7 | Failed | 41.2 | 44.1 |
| Contrastive + FedAvg | – | 31.1 | 32.4 | 31.8 | 41.1 | 44.1 | 46.4 |
| Proposed DCCO | 51.7 | 49.6 | 48.1 | 45.9 | 51.4 | 52.0 | 51.9 |
| Centralized CCO | | | | 52.6 | | | |
| Supervised from scratch | | | | 42.4 | | | |
| Full finetuning protocol (5K labeled training samples) | | | | | | | |
| CCO + FedAvg | – | Failed | 36.3 | 38.9 | Failed | 43.0 | 46.0 |
| Contrastive + FedAvg | – | 40.8 | 41.8 | 41.3 | 44.9 | 46.8 | 47.4 |
| Proposed DCCO | 51.5 | 50.5 | 49.3 | 47.7 | 51.5 | 52.1 | 52.0 |
| Centralized CCO | | | | 52.5 | | | |
| Supervised from scratch | | | | 42.4 | | | |
| Linear evaluation protocol (500 labeled training samples) | | | | | | | |
| CCO + FedAvg | – | Failed | 13.5 | 14.5 | Failed | 26.2 | 27.6 |
| Contrastive + FedAvg | – | 14.8 | 15.6 | 15.0 | 24.7 | 27.8 | 29.3 |
| Proposed DCCO | 33.3 | 31.2 | 29.5 | 26.7 | 33.3 | 33.5 | 33.5 |
| Centralized CCO | | | | 34.0 | | | |
| Supervised from scratch | | | | 13.1 | | | |

Table 1: Comparison of various approaches on CIFAR-100 dataset. Results for CCO + FedAvg and Contrastive + FedAvg are not reported when each client has only one sample, because we need at least two samples to compute these losses. Results for CCO + FedAvg are not reported when each client has four samples, because training with CCO loss based on four samples was unstable.

per client decreases and the number of clients per round increases, approaching the performance of centralized CCO training when each client has only one sample [3]. This is because the proposed DCCO approach optimizes a loss function based on aggregated global batch statistics and using more clients increases the diversity of samples in the global batch. However, we do not see such trend in the case of IID client datasets because each sample in every client dataset is already sampled in an IID fashion from the full CIFAR-100 dataset.

We do not report results for FedAvg variants of contrastive and CCO losses when each client has only one sample because we need at least two samples to compute these loss functions. The proposed DCCO approach can still be used in this case since it uses statistics aggregated from multiple clients to compute the loss. We do not report results for FedAvg variant of CCO loss when each client has only four samples because training was unstable in this case.

---

[3] Ideally, when each client has only one sample, the performance of DCCO with 512 clients per round should be close to the performance of centralized CCO with a batch size of 512. However, we notice a small ($\sim$1%) performance gap between the two approaches. We used the TensorFlow framework for running centralized training experiments and the TensorFlow Federated framework for running federated training experiments. This may account for the observed discrepancy. For example, the random data augmentation pipeline used during pretraining differs between these two setups in terms of whether they use stateful random operations or stateless random operations with explicitly provided seeds.

| Labeled data (Finetuning) | 1,524 cases | | | 3,057 cases | | | 4,592 cases | | |
|---|---|---|---|---|---|---|---|---|---|
| Num. clients (Pretraining) | 59,770 | | | 59,770 | | | 59,770 | | |
| Clients / Round | 64 | 128 | 256 | 64 | 128 | 256 | 64 | 128 | 256 |
| CCO + FedAvg | Failed | | | Failed | | | Failed | | |
| Contrastive + FedAvg | 38.9 | 41.8 | 43.9 | 45.3 | 46.7 | 47.8 | 47.7 | 49.1 | 51.5 |
| Proposed DCCO | 42.4 | 46.3 | 47.8 | 46.7 | 50.3 | 51.1 | 48.3 | 52.8 | 52.5 |
| Centralized CCO | 48.0 | | | 52.0 | | | 53.6 | | |
| Supervised from scratch | 30.6 | | | 35.9 | | | 39.2 | | |

Table 2: Comparison of various approaches on DERM dataset under full finetuning protocol. Results for CCO + FedAvg are not reported because training using within-client CCO loss was unstable as each client in this dataset has a maximum of six samples.

### 4.4.2 DERM Discussion

Table 2 shows the performance of various approaches under full finetuning protocol for different amounts of labeled data. The proposed DCCO approach clearly outperforms the FedAvg variant of contrastive loss-based training and achieves a performance close to that of centralized CCO training. The proposed approach also results in significant performance gains (13-17%) when compared to fully-supervised training from scratch demonstrating its effectiveness in leveraging unlabeled decentralized datasets. We do not report results for FedAvg variant of CCO loss because training was unstable in this case as each client in this dataset has a maximum of six images.

## 5 Related work

**Self-supervised learning** Contrastive methods, motivated by the *InfoMax* principle (Linsker, 1988), aim to maximize the mutual information between an instance and its encoded representation. Since a reliable estimate of mutual information is hard to obtain, methods often resort to optimizing the InfoNCE loss (van den Oord et al., 2018). Among contrastive methods, MoCo (He et al., 2020) uses a queue of examples to construct the contrastive pairs. It also uses a momentum encoder for training stability. MoCo v2 (Chen et al., 2020c) improves MoCo by adding a MLP projection head and more data augmentations. SimCLR (Chen et al., 2020a) removes the momentum encoder and uses the same network to encode both inputs. It also replaces the queue with large batch sizes. SimCLRv2 (Chen et al., 2020b) improves the original model by increasing the size of the encoder network and the depth of the non-linear projection module.

Different from contrastive learning, methods such as Barlow Twins (Zbontar et al., 2021) and VICReg (Bardes et al., 2022) define their objective functions based on certain statistics of the encodings such as cross correlation and variance. ARB (Zhang et al., 2022) learns representations by mapping the embeddings onto an orthonormal basis space. BYOL (Grill et al., 2020) and SimSiam (Chen & He, 2021) use asymmetric two-tower architectures that uses a small non-linear prediction network on one side and the stop-gradient operation on the other side. Tian et al. (2021) studies the effect of the non-linear predictor and stop-gradient in these methods and observe that both components are essential to prevent collapse.

Another line of self-supervised learning works use clustering for representation learning. DeepCluster (Caron et al., 2018) and SeLa (Asano et al., 2020) alternates between clustering in the encoding space and training the encoder network using a cluster prediction loss. SwAV (Caron et al., 2020) simultaneously clusters the data while enforcing consistency between cluster assignments produced for different augmentations of the same image. MSN (Assran et al., 2022) combines SwAV and BYOL while using masking to generate different views of the same image. Recently, several works (He et al., 2022; Fang et al., 2022; Wei et al., 2022; Chen et al., 2022; Xie et al., 2022; El-Nouby et al., 2021; Bao et al., 2021) also explored masked content prediction tasks for self-supervised representation learning.

**Federated dual encoding models** To enable large-batch contrastive training in federated settings, Wu et al. (2022b) and Zhang et al. (2020) propose to share individual data sample encodings between clients, raising privacy concerns. In contrast, our approach only shares encoding statistics aggregated across several clients. Zhuang et al. (2021; 2022) extend BYOL (Grill et al., 2020) to cross-silo federated settings by using a separate target encoder on each client. The online encoder on each client is updated by combining both local and global online encoders taking into account the divergence between them. He et al. (2021) extends SimSiam to cross-silo federated settings by using a separate personalized model on each client in addition to a common global model shared across clients. The global model is trained using the standard FedAvg strategy and the local models are trained to optimize the SimSiam loss function on local data while pushing the encodings generated by local models to be close to the encodings generated by the shared global model. Makhija et al. (2022) goes a step further and completely removes the shared global model. To enable collaborative learning, they use a public dataset and regularize client model training by forcing the encodings of the public dataset generated by each client model in a round to be close to the average of the encodings generated by all client models in the previous round. All these methods focus on cross-silo settings with small number of clients and thousands of samples per client. Makhija et al. (2022) showed that the performance of these methods degrades significantly as the number of clients increases. Different from all these approaches, we focus on decentralized datasets composed of a large number of small, non-IID client datasets.

Federated training of dual encoding models has also been explored in the context of recommendation systems where the goal is to predict next item from a sequence of previous items. In this setting, *sequence of previous items* and *next item* form the two inputs to a dual encoding model, and we learn a finite lookup table of item encodings while training a network that encodes sequences of item encodings. Ning et al. (2021) assumes that the entire item lookup table is accessible by each client and optimizes spreadout loss (Yu et al., 2020) on the item lookup table while maximizing the agreement between the encodings of previous items sequence and next item. Wu et al. (2022a) assumes that the full item lookup table is available only on the central server and proposes a strategy to sample items relevant for each client. The encodings of these items are transferred to the clients and used as negatives in contrastive loss-based training. Different from these works, we are focusing on continuous signals such as images where the concept of finite item lookup table is inapplicable.

## 6 CONCLUSION AND FUTURE WORK

In this work, we proposed an approach for training dual encoding models on decentralized datasets composed of a large number of small, non-IID client datasets. The proposed approach optimizes a loss function based on encoding statistics and simulates large-batch loss computation on individual clients by using encoding statistics aggregated across multiple clients. When each client participating in a training round performs only one local training step, each federated round of DCCO training is equivalent to a centralized training step on a batch consisting of all data samples participating in the corresponding round. Our experimental results on two image datasets show that the proposed approach outperforms FedAvg variants of within-client contrastive and CCO loss-based training. Our proposed approach also outperforms supervised training from scratch by a significant margin; demonstrating its effectiveness as a federated self-supervised learning approach.

This paper focused on self-supervised pretraining on unimodal datasets and the CCO loss function of Zbontar et al. (2021). In future work, we plan to experiment with multi-modal datasets, and evaluate the proposed aggregated statistics-based distributed learning strategy with other statistics-based loss functions such as Bardes et al. (2022). There are also several other interesting research directions we plan to pursue in the future. How can we extend the proposed approach to large client datasets? When training on large client datasets, we may need to perform multiple steps of local training on each client within one federated round. When we perform multiple training steps on a client, in each step, only a small subset of samples that are contributing to the loss function participate in gradient computation. Also, while the model weights change after each local step, the aggregated statistics used in the loss function remain constant. How can we address the effects of these *partial gradients* and *stale statistics*? The proposed DCCO approach uses two communication rounds between server and clients within one federated training round. How can we reduce this to one round of communication?

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

## A   PROOF OF EQUIVALENCE TO CENTRALIZED TRAINING

**Claim:** When we perform one step of local training on each participating client in a federated training round, one round of DCCO training is equivalent to one centralized training step on a batch composed of all data samples participating in the round.

**Proof:** Let $F_i^n$ and $G_j^n$ respectively denote the $i^{th}$ and $j^{th}$ components of encodings $F$ and $G$ of $n^{th}$ sample on a client. Based on the definition of $\langle F_i \rangle_C$, we get

$$\langle F_i \rangle_C = \langle F_i \rangle_k + StopGradient\left[\langle F_i \rangle_A - \langle F_i \rangle_k\right] \implies \frac{\partial \langle F_i \rangle_C}{\partial F_i^n} = \frac{\partial \langle F_i \rangle_k}{\partial F_i^n}. \tag{4}$$

Similarly, by definitions of $\langle (F_i)^2 \rangle_C, \langle G_j \rangle_C, \langle (G_j)^2 \rangle_C, \langle F_i G_j \rangle_C$, we get

$$\frac{\partial \langle (F_i)^2 \rangle_C}{\partial F_i^n} = \frac{\partial \langle (F_i)^2 \rangle_k}{\partial F_i^n}, \frac{\partial \langle G_j \rangle_C}{\partial G_j^n} = \frac{\partial \langle G_j \rangle_k}{\partial G_j^n}, \frac{\partial \langle (G_j)^2 \rangle_C}{\partial G_j^n} = \frac{\partial \langle (G_j)^2 \rangle_k}{\partial G_j^n},$$

$$\frac{\partial \langle F_i G_j \rangle_C}{\partial F_i^n} = \frac{\partial \langle F_i G_j \rangle_k}{\partial F_i^n}, \frac{\partial \langle F_i G_j \rangle_C}{\partial G_j^n} = \frac{\partial \langle F_i G_j \rangle_k}{\partial G_j^n} \tag{5}$$

By chain rule, we have

$$\frac{\partial \mathcal{L}_{CCO}}{\partial F_i^n} = \frac{\partial \mathcal{L}_{CCO}}{\partial \langle F_i \rangle_C} \frac{\partial \langle F_i \rangle_C}{\partial F_i^n} + \frac{\partial \mathcal{L}_{CCO}}{\partial \langle (F_i)^2 \rangle_C} \frac{\partial \langle (F_i)^2 \rangle_C}{\partial F_i^n} + \sum_{j=1}^{d} \frac{\partial \mathcal{L}_{CCO}}{\partial \langle F_i G_j \rangle_C} \frac{\partial \langle F_i G_j \rangle_C}{\partial F_i^n}$$

$$\frac{\partial \mathcal{L}_{CCO}}{\partial G_j^n} = \frac{\partial \mathcal{L}_{CCO}}{\partial \langle G_j \rangle_C} \frac{\partial \langle G_j \rangle_C}{\partial G_j^n} + \frac{\partial \mathcal{L}_{CCO}}{\partial \langle (G_j)^2 \rangle_C} \frac{\partial \langle (G_j)^2 \rangle_C}{\partial G_j^n} + \sum_{i=1}^{d} \frac{\partial \mathcal{L}_{CCO}}{\partial \langle F_i G_j \rangle_C} \frac{\partial \langle F_i G_j \rangle_C}{\partial G_j^n} \tag{6}$$

Substituting Eq. 4 and 5 in Eq. 6, we get

$$\frac{\partial \mathcal{L}_{CCO}}{\partial F_i^n} = \frac{\partial \mathcal{L}_{CCO}}{\partial \langle F_i \rangle_C} \frac{\partial \langle F_i \rangle_k}{\partial F_i^n} + \frac{\partial \mathcal{L}_{CCO}}{\partial \langle (F_i)^2 \rangle_C} \frac{\partial \langle (F_i)^2 \rangle_k}{\partial F_i^n} + \sum_{j=1}^{d} \frac{\partial \mathcal{L}_{CCO}}{\partial \langle F_i G_j \rangle_C} \frac{\partial \langle F_i G_j \rangle_k}{\partial F_i^n}$$

$$= \frac{1}{N_k} \frac{\partial \mathcal{L}_{CCO}}{\partial \langle F_i \rangle_C} + \frac{2}{N_k} \frac{\partial \mathcal{L}_{CCO}}{\partial \langle (F_i)^2 \rangle_C} F_i^n + \frac{1}{N_k} \sum_{j=1}^{d} \frac{\partial \mathcal{L}_{CCO}}{\partial \langle F_i G_j \rangle_C} G_j^n$$

$$\frac{\partial \mathcal{L}_{CCO}}{\partial G_j^n} = \frac{\partial \mathcal{L}_{CCO}}{\partial \langle G_j \rangle_C} \frac{\partial \langle G_j \rangle_k}{\partial G_j^n} + \frac{\partial \mathcal{L}_{CCO}}{\partial \langle (G_j)^2 \rangle_C} \frac{\partial \langle (G_j)^2 \rangle_k}{\partial G_j^n} + \sum_{i=1}^{d} \frac{\partial \mathcal{L}_{CCO}}{\partial \langle F_i G_j \rangle_C} \frac{\partial \langle F_i G_j \rangle_k}{\partial G_j^n}$$

$$= \frac{1}{N_k} \frac{\partial \mathcal{L}_{CCO}}{\partial \langle G_j \rangle_C} + \frac{2}{N_k} \frac{\partial \mathcal{L}_{CCO}}{\partial \langle (G_j)^2 \rangle_C} G_j^n + \frac{1}{N_k} \sum_{i=1}^{d} \frac{\partial \mathcal{L}_{CCO}}{\partial \langle F_i G_j \rangle_C} F_i^n \tag{7}$$

By chain rule, the gradients for encoding model parameters $\theta$ on $k^{th}$ client are given by

$$\left.\frac{\partial \mathcal{L}_{CCO}}{\partial \theta}\right|_k = \sum_{n=1}^{N_k} \sum_{i=1}^{d} \frac{\partial \mathcal{L}_{CCO}}{\partial F_i^n} \frac{\partial F_i^n}{\partial \theta} + \sum_{n=1}^{N_k} \sum_{j=1}^{d} \frac{\partial \mathcal{L}_{CCO}}{\partial G_j^n} \frac{\partial G_j^n}{\partial \theta}, \tag{8}$$

where $N_k$ is the number of samples on the client that contributed to $\mathcal{L}_{CCO}$.

Substituting Eq. 7 in Eq. 8, we get

$$\left.\frac{\partial \mathcal{L}_{CCO}}{\partial \theta}\right|_k = \frac{1}{N_k} \sum_{n=1}^{N_k} \sum_{i=1}^{d} \left( \frac{\partial \mathcal{L}_{CCO}}{\partial \langle F_i \rangle_C} + 2\frac{\partial \mathcal{L}_{CCO}}{\partial \langle (F_i)^2 \rangle_C} F_i^n + \sum_{j=1}^{d} \frac{\partial \mathcal{L}_{CCO}}{\partial \langle F_i G_j \rangle_C} G_j^n \right) \frac{\partial F_i^n}{\partial \theta} +$$

$$\frac{1}{N_k} \sum_{n=1}^{N_k} \sum_{j=1}^{d} \left( \frac{\partial \mathcal{L}_{CCO}}{\partial \langle G_j \rangle_C} + 2\frac{\partial \mathcal{L}_{CCO}}{\partial \langle (G_j)^2 \rangle_C} G_j^n + \sum_{i=1}^{d} \frac{\partial \mathcal{L}_{CCO}}{\partial \langle F_i G_j \rangle_C} F_i^n \right) \frac{\partial G_j^n}{\partial \theta} \tag{9}$$

The values of stats $\{\langle F_i \rangle_C\}_{i=1}^d, \{\langle (F_i)^2 \rangle_C\}_{i=1}^d, \{\langle G_j \rangle_C\}_{j=1}^d, \{\langle (G_j)^2 \rangle_C\}_{j=1}^d, \{\langle F_i G_j \rangle_C\}_{i,j=1}^d$ and loss $\mathcal{L}_{CCO}$ are same on all the clients participating in a DCCO training round. So, when each client performs one step of local training and the server averages the model updates from the clients by weighing them according to the number of contributing samples on each client, the equivalent global model gradient update is given by

$$
\begin{aligned}
\frac{\partial \mathcal{L}_{CCO}}{\partial \theta} &= \frac{1}{N} \sum_{k=1}^{K} N_k \frac{\partial \mathcal{L}_{CCO}}{\partial \theta} \bigg|_k \\
&= \frac{1}{N} \sum_{n=1}^{N} \sum_{i=1}^{d} \left( \frac{\partial \mathcal{L}_{CCO}}{\partial \langle F_i \rangle_C} + 2 \frac{\partial \mathcal{L}_{CCO}}{\partial \langle (F_i)^2 \rangle_C} F_i^n + \sum_{j=1}^{d} \frac{\partial \mathcal{L}_{CCO}}{\partial \langle F_i G_j \rangle_C} G_j^n \right) \frac{\partial F_i^n}{\partial \theta} + \\
&\quad \frac{1}{N} \sum_{n=1}^{N} \sum_{j=1}^{d} \left( \frac{\partial \mathcal{L}_{CCO}}{\partial \langle G_j \rangle_C} + 2 \frac{\partial \mathcal{L}_{CCO}}{\partial \langle (G_j)^2 \rangle_C} G_j^n + \sum_{i=1}^{d} \frac{\partial \mathcal{L}_{CCO}}{\partial \langle F_i G_j \rangle_C} F_i^n \right) \frac{\partial G_j^n}{\partial \theta} \quad (10)
\end{aligned}
$$

where $N$ is the total number of samples participating in the round.

If all the $N$ samples participating in a DCCO round are present on a single client, then according to Eq. 9, the gradients computed on that client using these $N$ samples will be same as the gradients in Eq. 10. Hence one round of federated DCCO training is equivalent to one step of centralized training on a batch composed of all $N$ samples participating in the federated round.

## B    HYPERPARAMETERS AND DATA AUGMENTATIONS

### CIFAR-100 DATASET

**Augmentations:** During self-supervised pretraining, we used all the data augmentations from Grill et al. (2020) except blur, and during supervised finetuning we only used flip augmentation.

**Federated pretraining:** For updating the model on server, we used Adam optimizer (Kingma & Ba, 2015) with cosine learning rate decay. We experimented with initial learning rates of $5e^{-3}$ and $1e^{-3}$ and report the best results. All models were trained for 100K rounds.

**Centralized pretraining:** We trained for 1000 epochs using Adam optimizer with batch size 512, initial learning rate of $5e^{-3}$, and cosine learning rate decay.

**Full network finetuning with 5K samples:** We trained for 100 steps using Adam optimizer with batch size 256, initial learning rate $5e^{-3}$, and cosine learning rate decay.

**Full network finetuning with 500 samples:** We trained for 40 steps using Adam optimizer with batch size 128, initial learning rate $5e^{-3}$, and cosine learning rate decay.

**Supervised training from scratch:** We trained for 100 epochs using Adam optimizer with batch size 256, initial learning rate 0.01, and cosine learning rate decay.

**Linear classifier training with 5K samples:** We trained for 1000 steps using LARS optimizer (You et al., 2017) with batch size 512, initial learning rate of 2.0, cosine learning rate decay, and a momentum of 0.9.

**Linear classifier training with 500 samples:** We trained for 400 steps using LARS optimizer with batch size 128, initial learning rate of 2.5, cosine learning rate decay, and a momentum of 0.9.

### DERM

**Augmentations:** During self-supervised pretraining, we used random rotation and all the data augmentations from Grill et al. (2020) except solarization. During supervised finetuning, we used all the data augmentations from Azizi et al. (2021).

**Federated pretraining:** For updating the model on server, we used LARS optimizer with cosine learning rate decay and a momentum of 0.9. For DCCO pretraining, the initial learning rate was set to 0.15, 0.9 and 1.8 while using 64, 128 and 256 clients per round, respectively. For FedAvg training

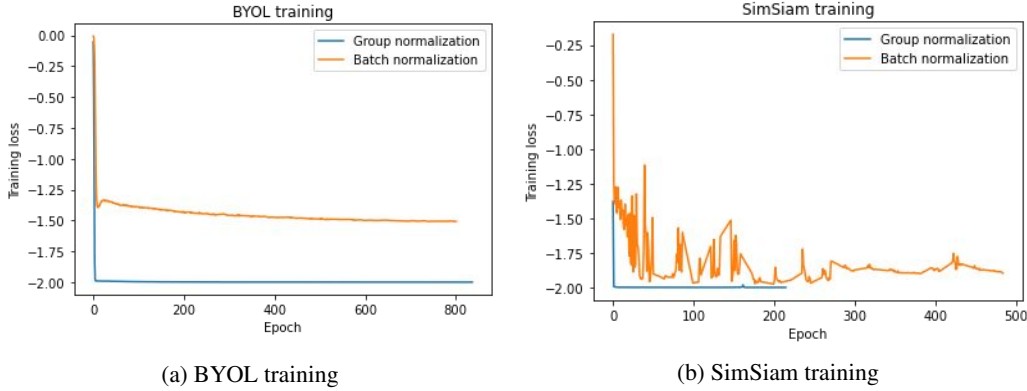

(a) BYOL training         (b) SimSiam training

Figure 3: ResNet14 training on CIFAR-100 dataset.

with contrastive loss, the initial learning rate was set to 0.15, 0.3 and 0.6 while using 64, 128 and 256 clients per round, respectively. All models were trained for 75K rounds.

**Centralized pretraining:** We trained for 320 epochs using LARS optimizer with batch size 512, initial learning rate of 0.6, cosine learning rate decay, and a momentum of 0.9

**Supervised training from scratch and full network finetuning:** We used Adam optimizer with a batch size of 128. For each experiment, we used the validation split of this dataset to search for the best learning rate (among $3e^{-4}$ and $1e^{-4}$) and number of training steps (up to 100k).

## C   BYOL AND SIMSIAM WITH GROUP NORMALIZATION

When we experimented with BYOL and SimSiam by replacing Batch Normalization (BN) with Group Normalization (GN), the models did not train well. Figure 3a shows the training loss when ResNet14 encoder is trained with BYOL approach on CIFAR-100 dataset. When we use GN instead of BN, the loss quickly drops close to its lowest possible value and the model does not learn after that. When we evaluated the BN and GN-based BYOL encoders after training them for 800 epochs with a batch size of 512, we achieved 41% and 3% accuracy, respectively, under linear evaluation protocol. This clearly shows that BN is crucial for BYOL to work well. Similar behavior was observed in the case of SimSiam (see Fig. 3b).

