# OpenReview forum: "Federated Training of Dual Encoding Models on Small Non-IID Client Datasets"
_ICLR.cc/2023/Conference — Submitted to ICLR 2023_

### Official Review · Reviewer_G6CA · 2022-10-23

**Confidence:** 3
**Correctness:** 3
**Technical Novelty And Significance:** 2
**Empirical Novelty And Significance:** 2
**Recommendation:** 5

**Clarity, Quality, Novelty And Reproducibility:**

The paper is easy to follow and clear, novelty is limited. With the experimental details provided in the paper, I believe it is producible.

**Strength And Weaknesses:**

Strengths is the proposed method outperforms the baseline methods by a large margin in terms of accuracy.

Weaknesses:
1. The limited novelty of the proposed method, it is simply a distributed way to calculate a loss function involving sum and product operations.

2. I may missed some details but I did not find comparison of communication of the proposed algorithm with federated learning baselines, the proposed algorithm does not utilize multi-step local training while FedAvg style algorithms use it, it is important to also compare the amount of communication when achieving the same accuracy, and the number of local steps of FedAvg style algorithms should be well-tuned.



**Summary Of The Paper:**

The paper proposes a distributed implementation of Cross Correlation Optimization (DCCO) loss for contrastive learning, experiments shows the proposed implementation of loss function outperforms some baseline algorithms in the federated learning setting.

**Summary Of The Review:**

Overall I feel the novelty is limited and some more metrics in experiments should be reported.

---

> ### Author Response · Authors · 2022-11-15
> **Thank you for your review.**
>
> We thank the reviewer for their review and address their concerns below.
>
> Simple distributed way to calculate loss: In this work, we have identified and demonstrated that for a certain family of SSL objective functions (statistics-based objectives such as CCO loss), it is possible to calculate and optimize large batch loss in federated settings without sharing individual client samples either in raw or encoded formats. We show that this is possible even when each individual client dataset is small. This is a significant result that is not shown in existing federated SSL works which focus on learning from a small number of stateful clients with large client datasets. This is the first work that presents a viable solution for federated self-supervised learning from a large collection of small stateless clients.
>
> Communication cost:  Compared to a standard FedAvg round, the proposed DCCO round performs additional communication for aggregating statistics. The number of statistical quantities communicated is O(d^2), where ‘d’ is the dimensionality of the encoding, which is significantly lower than the total model parameters. The proposed DCCO approach uses only one step of local training in each federated training round. We agree that this could increase the overall communication cost by increasing the total number of training rounds.  However, in a typical real-world FL setting (edge devices), the total number of participating devices is often very large and each device may be visited only a few times during the entire training process. So, the overall communication cost per device will still be low.
>
>
> Multi-step FedAvg training: We agree that it is common to use multiple steps of local training when FedAvg style approaches are used with large client datasets (which is the case in many existing works). However, this work focuses on small client datasets (for example a majority of clients in the DERM dataset have 2-4 images), and using multiple local steps on such small clients could lead to significant drift between client models and further slow down the convergence of FedAvg style algorithms. Hence, we use only one step of local training for the FedAvg baselines.

---

> ### Author Response · Authors · 2022-11-29
> **Gentle Reminder**
>
> Dear reviewer,
>
> Thank you again for your time reviewing our paper. We would appreciate it if you could confirm that our responses address your concerns. We would also be happy to engage in further discussions to address any other questions that you might have.
>
> Best regards,
>
> Paper3004 Authors

---

### Official Review · Reviewer_j562 · 2022-10-25

**Confidence:** 2
**Correctness:** 3
**Technical Novelty And Significance:** 3
**Empirical Novelty And Significance:** 3
**Recommendation:** 6

**Clarity, Quality, Novelty And Reproducibility:**

The motivation of the problem is clearly described in the introduction, and the proposed approach is easy to understand and well supported by texts and graphs. The paper focuses on a novel scenario of training dual encoding models, which is challenging since the existing methods perform poorly. The experiment settings are detailed, but there is no code in the supplementary materials.

**Strength And Weaknesses:**

**Strengths:**
1. This paper provides a detailed description of the motivation. The authors discover the limitations of the existing methods and solve the problems under the limited environment that is closer to reality, thus making this work challenging and meaningful.
2. The details of the proposed approach are well supported by texts and graphs, and the experiment settings are proper for proving the advantages of the approach.

**Weaknesses:**
1. Figure 2 presents the overview of the approach with separate graphs. However, the corresponding relations of the two graphs are vague. Figure 2 can be replaced by a single detailed graph of DCCO and explain the relations in the caption.
2. For the proposed method, the authors should give the algorithm for readers to understand better.
3. We would like to see the influence of other environmental parameters on experimental performance, such as $\lambda$.


**Summary Of The Paper:**

This paper focuses on the problem of training dual encoding models on decentralized datasets, which has few existing works. Moreover, the authors consider a challenging scenario that each client possesses a small and non-IID dataset, where directly utilizing the existing centralized methods decreases efficacy. Hence, the authors propose a novel method, Distributed Cross Correlation Optimization (DCCO), to train dual encoding models on decentralized datasets with federated learning. This paper provides several experimental results and proofs to show the advantages of DCCO.

**Summary Of The Review:**

This paper proposes a novel method to solve the problem of training dual encoding models on decentralized datasets and proves its advantages on small and non-IID datasets over existing centralized methods. The details of the problem and the methods are clear and easy to understand for readers. However, there are still some problems that affect the fluency of the paper. The figures and the experiments should be supplemented to help readers grasp the idea and prove the generalizability of the approach. Overall, this work is meaningful in the area of dual encoding model training and can be further studied and optimized.

---

> ### Author Response · Authors · 2022-11-15
> **Thank you for your review.**
>
> We thank the reviewer for providing a fair assessment of our contributions.  Hopefully, our discussion addresses their remaining concerns.
>
> Figure 2:  We apologize for any lack of clarity in Figure 2.  Our intention in separating Fig.2 into two subfigures was to use one figure to provide a high-level overview of a DCCO training round and use another one to explain the client-side training in detail. The bottom figure corresponds to the second half of the DCCO round in which the clients receive aggregated statistics and compute model updates. We will make this figure better in the final draft based on the reviewer's suggestions.
>
> Algorithm: Thank you for the constructive suggestion. We will add a table detailing the algorithm in the final version.
>
> Effect of Lambda: We do think that studying the effect of this parameter in detail is interesting. We did evaluate a few different values (0, 10, 20, 30, 40) for this parameter in some early experiments and found that 20 is a good value. However, the performance difference between these values was not significant (less than 2%).

---

> > ### Comment · Reviewer_j562 · 2022-11-26
> > **Response to the Authors**
> >
> > Dear authors,
> >
> > Thank you for providing a response to my questions.  I have read the concerns/issues of other reviewers, the authors claim DCCO  can use Secure Aggregation, which needs more descriptions.

---

> > > ### Author Response · Authors · 2022-11-29
> > > **Secure Aggregation**
> > >
> > > Thank you for your question surrounding Secure Aggregation and how it can be used with DCCO.  Secure aggregation [1]  is simply a class of secure multi-party computation algorithms wherein a group of mutually distrustful parties (the clients and the central server) collaborate to perform aggregation of their values (typically variable updates, but embedding statistics in DCCO) without revealing to each other any information other than what may be learned from the aggregated value.  By performing aggregation in this way, the server is unable to infer the exact values contributed by any given client, but the resulting value computed on the server is equivalent to the intended output of the aggregating function (addition in our case).  So by using secure aggregation on client embedding statistics, the resulting tensor is equivalent to the sum over all of the clients (the global embedding statistics), but the embedding statistics from any particular client cannot be inferred by the server.  It is thus trivially compatible with algorithms such as ours because it is just a means of keeping the communication between the clients and the server cryptographically secure, and preventing an honest-but-curious server from inferring individual client statistics.
> > >
> > > References
> > >
> > > [1] Bonawitz, K. et al., Practical Secure Aggregation for Federated Learning on User-Held Data, https://arxiv.org/abs/1611.04482

---

### Official Review · Reviewer_RUBX · 2022-10-27

**Confidence:** 5
**Correctness:** 2
**Technical Novelty And Significance:** 2
**Empirical Novelty And Significance:** 2
**Recommendation:** 3

**Clarity, Quality, Novelty And Reproducibility:**

Clarity: The paper’s content organisation could be improved.

Quality: The paper’s overall quality is below the threshold.

Novelty: The novelty is fair. But the overall contribution to the FL community is limited.

Originality: The paper is an incremental work.


**Strength And Weaknesses:**

Strengths

1. The target problem is meaningful in real-world applications for FL.


 Weaknesses

1. This paper’s contribution is incremental.

2. Too much data sharing may lead to security and privacy concern, which is not discussed in the paper. Communication efficiency is another problem;

3. Eq.2 is not bounded between 0 and 1 when local stats are globally aggregated.

4. One step of FL equivalent to central training will establish for almost all optimization methods, which can not prove the advantageousness of DCCO.

5. The work employed the local loss function of Barlow Twins, but replaced the calculation of C_ij (Eq.2) with correlation coefficient term. How can this replacement lead to better performance? Discussion and ablation experiments are needed.

6. Lack of experiments comparison with other self-supervised FL methods such as [1][2].

[1] Divergence-aware Federated Self-Supervised Learning

[2] Towards Communication-Efficient and Privacy-Preserving Federated Representation Learning


**Summary Of The Paper:**

In this paper, the authors propose a domain-aware representation learning method (FedDAR) for the non-iid FL problem. The FedDAR assumes data on clients are from multiple domains and learns a classifier head for each domain. A representation module is shared for all classifier heads and updated by the vanilla FedAvg. The authors also proposed a second-order aggregation method to update domain-aware classifier heads, whose effectiveness is validated by ablation studies. Experiments on both synthetic data and real-world datasets validate the effectiveness of the proposed method.



**Summary Of The Review:**

Please refer to the comments in Strengths and Weaknesses.

---

> ### Author Response · Authors · 2022-11-15
> **Thank you for your review.**
>
> We thank the reviewer for their review listing specific actionable items they found the paper was lacking.  Hopefully, the following discussion helps to alleviate some of these concerns.
>
> Incremental contribution: We respectfully disagree with the reviewer on this point. This work focuses on federated SSL using a large number of small client datasets, which is different from all existing federated SSL works which focus on a small number of large clients each of which consists of thousands of samples. When client datasets are large, pretty much all existing SSL methods could be made to train successfully with naive adaptations. However, when the client datasets are small, combining existing SSL methods with FedAvg is not effective because the local training objectives are poor approximations of the global objective. Our inability to train BYOL and SimSiam (see Appendix C), and poor experimental results for ‘CCO + FedAvg’ and ‘Contrastive + FedAvg’ confirm this hypothesis. In this work, we introduce DCCO which facilitates the computation of large-batch CCO loss, and its gradient, in a distributed manner. We prove that the proposed way of computing loss and gradients implies an equivalence between DCCO and centralized CCO, and experimentally show that the proposed DCCO is effective even when individual client datasets are small.
>
> Local Barlow twins loss replacing C_ij with cross-correlation: If we naively adapt Zbontar et al. (2021) to the federated setting using FedAvg, then the CCO loss on each client uses cross-correlation coefficients C_ij that are computed using local statistics. In contrast to this, the proposed DCCO approach computes CCO loss on each client using the C_ij computed from aggregated large batch statistics. Our experimental results clearly show that the proposed DCCO approach significantly outperforms ‘local stats-based CCO + FedAvg’. This is because the average of CCO losses computed on several small batches is not equal to the CCO loss on a large batch obtained by combining the small batches (which is different from standard loss functions such as cross-entropy loss or mean squared error loss for which batch loss is equal to the average of individual sample losses). Note that the statistics <F_i>, <F_i^2>, <G_j> <G_j^2> <F_iG_j> used to compute C_ij are sample averages and hence large batch statistics can be obtained by simply averaging small local batch statistics. Hence, we first average local statistics to obtain large-batch statistics and then use large-batch statistics to compute C_ij and CCO loss. As shown in the paper, this results in equivalence between federated DCCO and centralized CCO training.
>
> Privacy concerns:  Note that the server aggregates the embedding statistics across several clients before sharing them back with the clients. So individual clients will never have access to the statistics of a single client. We would like to point out that the proposed statistics aggregation can be combined with methods like SecAgg, DP, or trusted computing to maintain client privacy during aggregation.
>
> Communication cost: Compared to a standard FedAvg round, the proposed DCCO round performs additional communication for aggregating statistics. The number of statistical quantities communicated is O(d^2), where ‘d’ is the dimensionality of the encoding, which is significantly lower than the total model parameters.
>
> Eq.2 is not bounded between 0 and 1: Note that the quantities C_ij are correlation coefficients and hence are always bounded between -1 and +1. This is the case irrespective of whether the statistics <F_i>, <F_i^2>, <G_j> <G_j^2> <F_iG_j> are local batch-based or aggregated.
>
> One step of FL training is always equal to one step of centralized training: We respectfully disagree with the reviewer on this point. One step of FL training is equal to centralized training only for batch-insensitive loss functions, i.e., loss functions for which the loss value for a single sample does not depend on other samples in the batch. This is the case for standard loss functions such as cross-entropy and mean squared error which are commonly used for classification and regression tasks. For batch-sensitive loss functions such as contrastive loss or the CCO loss, the average of losses computed on several small batches is not equal to the loss on a large batch obtained by combining the small batches. Hence, for these losses, one step of FL training is not equal to one step of centralized training. This is also demonstrated experimentally by the performance gap between DCCO and ‘CCO+FedAvg’ and ‘Contrastive + FedAvg’.
>
> Comparison to alternative approaches: Note that this work focuses on federated SSL with a large number of small clients. The approaches pointed out by the reviewer are not applicable to this setting since they are designed to learn from a small number of stateful clients (with personalized/local models).

---

> > ### Comment · Reviewer_RUBX · 2022-12-12
> > **Not convinced regarding novelty and part of the technical design.**
> >
> > Thanks for the authors' response. However, I am not fully convinced regarding the novelty and part of the technical design. Therefore, I still stand by my original scores.

---

> ### Author Response · Authors · 2022-11-29
> **Gentle Reminder**
>
> Dear reviewer,
>
> Thank you again for taking the time to review our paper.  We would appreciate it if you could conform that our responses address your concerns.  In particular, we have tried to clarify several misunderstandings about our method which may have caused a significant impact on your evaluation of our paper.  We believe that we have clarified this confusion and hope that you could take the time to re-evaluate our paper.  We would also appreciate any further feedback you could provide which might improve future revisions of the paper.
>
> Best regards,
>
> Paper3004 Authors

---

### Official Review · Reviewer_xR7P · 2022-10-28

**Confidence:** 5
**Correctness:** 2
**Technical Novelty And Significance:** 1
**Empirical Novelty And Significance:** 1
**Recommendation:** 3

**Clarity, Quality, Novelty And Reproducibility:**

The novelty is limited. The proposed approach is a simple adaptation of the existing work in Zbontar et al. (2021). No code and the Dermatology dataset are provided; therefore, reproducibility is limited. In short, there is room for improvement, and the paper does not advance my knowledge in the field.

**Strength And Weaknesses:**

The problem studied in this paper is interesting. However, the paper has serval major weaknesses as follows:

- The paper leverage privacy concerns to motivate the work using federated learning (no sharing data). However, the proposed approach shares statistical information from encodings, significantly increasing the privacy risk of the client's local data. Note that federated learning does not automatically offer privacy protection since the server can extract clients' local data from the shared information. Therefore, the motivation for the work is unclear.

- The proposed approach is a trivial adaptation of the existing CCO loss used for training dual encoding models in Zbontar et al. (2021). The technical contribution is not significant. What are fundamental questions or challenging problems the paper aims to address?

- Experimental results are unconvincing. The datasets used are small, and the model utility is shallow (<50\% in most cases). The practicability of the proposed approach and setting is unconvincing. Who is going to use models with such utility? In addition, the comparison is unfair. The proposed approach used significantly larger data to train the model than a minimal dataset to train the centralized model. That does not show the advantage of the proposed approach. Why don't we gather all the data to train a usable centralized model instead of sticking with a poorly federated learning model? Again, the privacy concerns are invalid in this setting since the proposed approach does not offer any privacy protection. What is the cost of the proposed method in terms of communication and computation? How do the encodings improve after and before using the proposed approach? The paper does not shed light on understanding the core contribution of the proposed approach. Why FedAvg? There are better aggregations to use. That highlights a poor treatment for critical components, such as privacy, model/data utility, practicability, and usability, of the proposed approach given the learning setting.

**Summary Of The Paper:**

This paper presents an algorithm to train dual encoding models in federated learning. The key idea is sharing aggregated encoding statistical information across clients. Experiments were conducted on two benchmark datasets, including CIFAR-100 and Dermatology. The Dermatology dataset consists of de-identified images of skin conditions captured using consumer-grade digital cameras. The results show that the proposed algorithm achieves better utility than some baselines.

**Summary Of The Review:**

This paper has limited contribution. The motivation of the work is unclear; the proposed approach is simple without offering a deep understanding of how it works. Experimental results are unconvincing, and a thorough evaluation and analysis are needed.

---

> ### Author Response · Authors · 2022-11-15
> **Thank you for your thoughtful comments.**
>
> We are grateful to the reviewer for taking the time to read through our work, and for pointing out the parts that need further clarification. We are optimistic that our discussion will help them to better understand our core contributions and cause them to provide a more accurate assessment of our work. We address each of their main concerns below.
>
> Sharing statistical information: Our statistics aggregation procedure is similar (from a privacy standpoint) to standard FL weight aggregation procedure and thus may contain the same vulnerabilities. We would like to point out that there are ways to perform this aggregation while maintaining client privacy.  In particular, the proposed statistics aggregation can be performed with Secure Aggregation (e.g. https://eprint.iacr.org/2017/281) to prevent an honest but curious server from learning individual client data. This work focuses on enabling large-batch CCO loss computation in federated settings by reconfiguring the loss as a set of client-specific calculations that can be securely aggregated, rather than providing a new way to do secure aggregation.
>
> Trivial adaptation:  We would like to emphasize that the proposed approach is not a trivial adaptation of Zbontar et al. (2021). One of the alternative approaches presented in our experiments (referred to as CCO + FedAvg) is a trivial adaptation of Zbontar et al. (2021) to federated settings. We show that doing this trivial adaptation causes downstream task performance to degrade significantly (up to 20% relative to centralized CCO). This is because CCO loss on a large batch is not equivalent to the average of CCO losses computed on several small batches that make up the large batch (which is different from standard loss functions such as cross-entropy loss or mean squared error loss for which batch loss is equal to the average of individual sample losses). We introduce a non-trivial extension of the CCO loss computation, which facilitates the computation of large-batch loss, and its gradient, in a distributed manner (and in such a way that we can use Secure Aggregation). We prove that the proposed way of computing loss and gradients implies an equivalence between DCCO and centralized CCO. We further verify our theoretical result experimentally and show that we can successfully pretrain our models on a distributed population of clients.
>
> Unfair comparison: We would like to emphasize that the focus of this paper is on limited labeled data settings, i.e., leveraging unlabeled data to improve the performance when labeled data is limited. Hence, the datasets used in our experiments are comprised of a small amount of labeled data and a relatively large amount of unlabeled data. Note that we already compare the proposed DDCO approach to a centralized training approach (referred to as centralized CCO) that uses both labeled and unlabeled data. Our results show that the proposed approach gives competitive performance when compared to centralized CCO without the need for transfering client data to a central server. We provide results for a model trained only on labeled data (referred to as ‘Supervised from scratch’) to show the effectiveness of leveraging unlabeled data.
>
> Small datasets and shallow utility: Our experimental results clearly show that the proposed DCCO approach outperforms trivial federated adaptations of recent state-of-the-art self-supervised learning approaches such as contrastive learning and Zbontar et al. (2021) by a significant margin when the same dataset and model are used for training. The proposed DCCO approach also performs competitively when compared to ‘Centralized CCO’ which transfers the entire unlabeled and labeled data to a central server and performs centralized training. While absolute performances could be improved by using larger neural networks, achieving state-of-the-art results on a particular dataset is not the goal of this paper. We would also like to point out that CIFAR-100 is a standard benchmark in Federated SSL literature (see the DAFSSL or the FedU paper for examples).
>
> Why FedAvg?: Note that the proposed training strategy is agnostic to the server optimizer used. While the weight updates are simply averaged, we use sophisticated optimizers such as Adam (CIFAR-100) and LARS (DERM) as the server optimizers. All the optimization hyperparameters are presented in the Appendix.
>
> Communication cost: Compared to a standard FedAvg round, the proposed DCCO round performs additional communication for aggregating statistics. The number of statistical quantities communicated is O(d^2), where ‘d’ is the dimensionality of the encoding, which is significantly lower than the total model parameters.
>
> Computation cost: Compared to a standard FedAvg round, the proposed DCCO round performs forward pass twice, once for computing the statistics and once during gradient computation. Backpropagation is run only once. So the total computational cost goes up by 1.5x approximately.

---

> ### Author Response · Authors · 2022-11-29
> **Gentle Reminder**
>
> Dear reviewer,
>
> Thank you again for taking the time to review our paper. We would appreciate it if you could conform that our responses address your concerns. In our rebuttal, we have tried to address several misunderstandings about our method which may have impacted your evaluation of our paper. We believe that our responses have provided clarity on this confusion and were hoping that you could take the time to re-evaluate our paper.  We would also appreciate any further feedback you could provide which could improve further revisions of the paper.
>
> Best regards,
>
> Paper3004 Authors

---

### Decision · Program_Chairs · 2023-01-20

**Decision:**

Reject

**Justification For Why Not Higher Score:**

limited technical novelty

**Justification For Why Not Lower Score:**

n/a

**Metareview: Summary, Strengths And Weaknesses:**

This paper proposes a method to perform self supervised learning through dual encoding in an FL environment with many clients with a small number of non-iid data.

Three reviewers suggested reject, particularly on the grounds of limited technical novelty, while one reviewer recommended accept. Although no consensus was reached even during the discussion period, positive reviewer did not strongly support accept.

This AC also carefully reviewed this paper, but agreed with the opinions of the reviewers who argued for rejection. I understand considering the scenario where each client has a small number of data samples, but in the main experiment, the fact that the number of clients is extremely large and each client only has a maximum of 16 samples is not very persuasive. In addition, more reliable verification of privacy leak issues according to additionally shared information is required.